# Cost-Effective Optical Wireless Sensor Networks: Enhancing Detection of Sub-Pixel Transmitters in Camera-Based Communications

**DOI:** 10.3390/s24103249

**Published:** 2024-05-20

**Authors:** Idaira Rodríguez-Yánez, Víctor Guerra, José Rabadán, Rafael Pérez-Jiménez

**Affiliations:** 1Institute for Technological Development and Innovation in Communications (IDeTIC), Universidad de Las Palmas de Gran Canaria (ULPGC), 35017 Las Palmas de Gran Canaria, Spain; idaira.rodriguez119@alu.ulpgc.es (I.R.-Y.); jose.rabadan@ulpgc.es (J.R.); 2Pi Lighting Sarl, 1950 Sion, Switzerland; victor.guerra@pi-lighting.com

**Keywords:** wireless sensor networks, optical camera communication, sub-pixel

## Abstract

In the domain of the Internet of Things (IoT), Optical Camera Communication (OCC) has garnered significant attention. This wireless technology employs solid-state lamps as transmitters and image sensors as receivers, offering a promising avenue for reducing energy costs and simplifying electronics. Moreover, image sensors are prevalent in various applications today, enabling dual functionality: recording and communication. However, a challenge arises when optical transmitters are not in close proximity to the camera, leading to sub-pixel projections on the image sensor and introducing strong channel dependence. Previous approaches, such as modifying camera optics or adjusting image sensor parameters, not only limited the camera’s utility for purposes beyond communication but also made it challenging to accommodate multiple transmitters. In this paper, a novel sub-pixel optical transmitter discovery algorithm that overcomes these limitations is presented. This algorithm enables the use of OCC in scenarios with static transmitters and receivers without the need for camera modifications. This allows increasing the number of transmitters in a given scenario and alleviates the proximity and size limitations of the transmitters. Implemented in Python with multiprocessing programming schemes for efficiency, the algorithm achieved a 100% detection rate in nighttime scenarios, while there was a 89% detection rate indoors and a 72% rate outdoors during daylight. Detection rates were strongly influenced by varying transmitter types and lighting conditions. False positives remained minimal, and processing times were consistently under 1 s. With these results, the algorithm is considered suitable for export as a web service or as an intermediary component for data conversion into other network technologies.

## 1. Introduction

### 1.1. Optical Camera Communications

An unprecedented increase in the implementation of sensor networks for a variety of applications, such as air quality monitoring or smart energy management, has occurred in the last few years. These sensor networks generate vast amounts of data that need to be efficiently and reliably transmitted for analysis and action. Traditional wireless communication technologies like WiFi and Bluetooth have been widely used in these fields. However, these technologies often encounter problems related to spectrum congestion, electromagnetic interference, costs, energy efficiency and limitations in the ability to connect a large number of devices simultaneously [1].

Conversely, Optical Wireless Communications (OWC) technology offers significant advantages, including robust bandwidth, minimal latency, cost-effective installation, and operational efficiency. The interest of both the scientific and industrial communities on OWC has been reflected on the standardization efforts of the last years, e.g., IEEE 802.15.7r1, IEEE 802.11bb, and ITU recommendation G.9991 [2]. In these standards, the focus extends to the two primary types of receivers commonly used in OWC: photodetectors and image sensors. On the one hand, photodetectors offer higher speeds, making them suitable for demanding applications. On the other hand, image sensors provide cost-effective solutions and spatial separation capabilities while maintaining the recording functionality. This last technology is known as OCC.

OWC spans the electromagnetic spectrum from infrared to ultraviolet. However, it is the visible range that attracts the most interest due to the significant advancements in industrialization both in terms of transmitters and potential receivers. This technology is commonly referred to as Visible Light Communication (VLC). Consequently, there is a substantially broader range of publications in this area compared to the non-visible spectrum. This is particularly evident in fields related to sensor networks and IoT, driven by the low costs associated with advancements mentioned in commercialization and industrialization, as can be seen for example in [3]. The emphasis on the visible spectrum is also observed in OCC for similar reasons. Nevertheless, it is noteworthy that many cameras also operate within the infrared spectrum, so this range can also be exploited at low cost and with less visual impact in the scenarios where communication systems are installed.

In the domain of OCC, two acquisition modes are usually available among commercial image sensors, as also detailed in the aforementioned standards: Global Shutter (GS) and Rolling Shutter (RS). Whilst GS utilizes Charge-Coupled Device (CCD) sensors, enabling the simultaneous capture of all pixels, Rolling Shutter (RS) systems, primarily based on Complementary Metal-Oxide-Semiconductor (CMOS) technology, capture pixels sequentially rather than all at once, processing the images row by row. Although each sensor type has its own specific advantages, CMOS sensors are the most prevalent due to their cost-effectiveness and versatility across a wide range of applications.

OCC finds its importance in the myriad of applications it offers. The spatial separation of light sources thanks to the use of image-forming optics, combined with its low cost and ease of installation, makes it an optimal choice for applications with a high density of transmitters, such as Wireless Sensor Networks (WSNs) [4], the Internet of Things (IoT) [5], Smart and Cognitive Cities [6], and Industry 4.0 [7]. Furthermore, OCC excels in applications that require high precision, such as Visible Light Positioning (VLP), with achievable accuracy on the order of centimeters [8]. It makes it also fit well with object localization tasks [9]. As a result, it also plays a vital role in Intelligent Transportation Systems (ITSs) as part of Smart Cities, where it enhances communication between autonomous vehicles and infrastructure elements such as traffic lights, street lights, and other vehicles, ultimately improving the overall system performance [10]. Figure 1 depicts a conceptual illustration on the use of OCC in the aforementioned scenario.

However, in the various application domains where OCC may be a suitable option, the transmitting elements may not always be situated at such a close distance to the camera or have such large dimensions so as to occupy a significant proportion of the image pixels. This situation is intriguing for several reasons: using small transmitters can lead to energy consumption reduction, placing transmitters at distant locations can greatly expand coverage areas, and having a small projection of the transmitter on the image sensor allows increasing the number of transmitters detectable by a single camera. Consequently, it is highly likely that their proportion will be less than a few pixels, making the specific study of this scenario of great interest. The sub-pixel scenario is defined as one in which the projection of the transmitter onto the image sensor is less than one pixel [4].

It is important to consider that even if the pixel’s projection on the image sensor is less than one pixel, the light projection onto it may occupy a larger number of sensor cells. This depends on various factors such as the average brightness of the scene, the brightness and directivity of the source, or distortions introduced by the channel. A higher average scene brightness leads to less light from the source being captured as it is overshadowed by ambient light. A brighter source increases the likelihood of its emission affecting multiple pixels due to channel-induced dispersion.

Imperfections in the optical system are also responsible for a source occupying more than one pixel on the image sensor even if its projection is less than one pixel. This occurs because when light from a point source passes through a lens or optical system, it undergoes modifications and dispersion due to aberrations in the lens and characteristics of the optical system. This results in an extension or spreading of the original point in the projected image. To describe how an optical system or lens defocuses a point of light in an image, the Point Spread Function (PSF), also known as the Potential Spread Function, is generally used [4]. Typically, it is represented as a two-dimensional function, where each point in the output image is associated with a value indicating the light intensity at an input point and the volume of the illuminated region.

In the domain of OCC, transmitter discovery tasks have received limited attention in the broader scope of OCC research. Specifically, the exploration of transmitter discovery under sub-pixel conditions has been virtually non-existent. This dearth of research not only hinders the broader application of OCC but also restricts its utility when dealing with distant or low-light transmitters. In this paper, an algorithm is presented to address this challenge, enabling the discovery of transmitters in OCC systems where transmitters and receivers remain stationary even under such demanding conditions. The approach leverages both luminance parameters and the defined frame structure, providing robust transmitter detection. The algorithm is evaluated across various scenarios and lighting conditions.

The remainder of this paper is structured as follows. To provide context, the OCC general literature is reviewed, and the sub-pixel scenario is defined in Section 1.1. In Section 1.2, prior research related to discovery algorithms in OCC and the sub-pixel scenario works is delved into. In Section 1.3, the impact of OCC on 6G is discussed. The software developed is detailed in Section 2, while Section 3 outlines the experimental design and key metrics of interest. Finally, the obtained results are presented in Section 4 and the conclusions drawn from this work are discussed in Section 5.

### 1.2. Related Works

Sub-pixel OCC scenarios have not been extensively studied yet in the literature. Nevertheless, they have been indirectly addressed on several works. For instance, in [11], two equations which facilitate the determination of the 2D pixel projection (*x*, *y*) occupied by a transmitter on an image sensor, (Equation 1) and (2), are formulated and experimentally validated. These equations facilitate the determination (*x*, *y*) occupied by a transmitter on an image sensor. These values are determined based on the separation (*D*) between the object and the camera as well as the distances (dx, dy) and the camera’s field of view (FOVx, FOVy) relative to the transmitter along with the camera’s pixel resolution (Nx, Ny).
(1)$$\begin{array}{cc} \;x & =\frac{2\cdot Nx\cdot \arctan\left(\frac{dx}{2D}\right)}{FOVx} \end{array}$$
(2)$$\begin{array}{cc} \;y & =\frac{2\cdot Ny\cdot \arctan\left(\frac{dy}{2D}\right)}{FOVy} \end{array}$$

There have also been several studies that have explored long-distance OCC links. However, these studies involved modifications to the camera to increase the transmitter’s projections onto the image sensor, spanning multiple pixels. For instance, in [12], a novel technique was developed to extend the reach of RS-based OCC, achieving a link distance of 400 m through the use of telephoto lenses. In [6], another RS link spanning 328 m was implemented, where the LED array was detected using a magnifying lens with a sufficiently narrow horizontal field of view (28.1 degrees) to ensure the transmitter’s projection always exceeded 2 × 2 pixels. In [13], sensor image parameters were adjusted to blur the transmitter’s projection on the image sensor, thereby increasing the RS link distance.

Being able to work with transmitter projections at or below a single pixel on the receiver without the need for modifications is of significant interest. The use of telephoto or magnifying lenses eliminates the need for cameras primarily used for recording purposes and confines the spatial separation of transmitters to a small region. Consequently, working with sub-pixel projections enables both the extension of link distances and an increase in the quantity of such links within the same environment. However, the number of studies that have genuinely addressed these conditions is rather limited, often focusing on verifying the feasibility of such communications. For instance, in [14], the feasibility of a GS link is demonstrated, wherein the transmitter’s projection onto the receiver is sub-pixel, which is facilitated by the fact that the emitted light energy from the LED affects multiple pixels. In [4], this concept is further tested, this time under adverse weather conditions, using an RS link.

Several studies have addressed both transmitter detection and tracking tasks in OCC systems. However, none of them specifically tackle sub-pixel scenarios, making it the sole algorithm capable of universally addressing this aspect. Other approaches focus on transmitters with a predefined size and shape or cater to specific applications, rendering them challenging to extrapolate. For instance, in [15], segmentation and edge detection techniques are employed to detect LED arrays at close distances. In [13], a method for transmitter detection and tracking in mines is defined with the sole distinguishing parameter being the transmitter’s luminosity. In [9], a method for tag detection based on OCC technology is proposed and tested for object identification. In this case, it does not involve communication frames but rather the constant transmission of identifiers to locate objects. The detection process involves resizing images to VGA format and applying space–time correlation based on defined tags.

Furthermore, these algorithms are hardly suitable for use in sensor networks beyond the sub-pixel context. For example, in [16], if the transmitter projections are not sufficiently high, there is a high likelihood of losing their information when converting the image to VGA format. This limits the quantity of transmitters that can be deployed in scenarios and the maximum link distances. Additionally, working with communication frames instead of a few identifiers would significantly increase the computational and temporal costs of the algorithm. In [15], dependence on transmitter shape and size would also restrict its use in dense sensor networks. Finally, ref. [13] is only viable in dark scenarios without the presence of other light sources. All these factors underscore the fundamental importance of the method proposed in this document for the application of OCC in sensor networks. It contributes to the development of more cost-effective sensor networks with reduced acquisition and energy costs, simplified installation, and minimal impact on the existing radio spectrum.

### 1.3. OCC as an Enabler for 6G

As the deployment of 5G progresses, it becomes paramount to establish the groundwork for the subsequent generation, 6G. This forthcoming phase will enable cutting-edge technologies. For instance, consider realistic holographic communication, which enhances the natural perception of 3D holograms, or Extended Reality (ER), amalgamating augmented, virtual, and mixed reality to offer high-quality immersive experiences. The Tactile Internet, characterized by low latency, enabling real-time applications ranging from remote surgery to industrial control, is also noteworthy. In addition, multisensory experiences encompassing all human senses, including taste and smell, open up new possibilities across various industries. Furthermore, detailed digital twins of physical objects advance automation and intelligence in manufacturing. Similar to its role in 5G, the Internet of Things (IoT) remains significant, enabling broader and faster data acquisition. Additionally, intelligent transport and logistics ensure secure and efficient mobility [9]. These innovations signify an important shift in how technology is interacted with and hold the promise of revolutionizing industries such as healthcare, terrestrial and aerial vehicular networks, satellite communications, entertainment, logistics, and industrial settings.

To facilitate the proliferation of these new technologies and advancements, it is essential to ensure a series of changes in communication technologies. These requirements primarily encompass aspects related to latency, data rates, connection density, mobility, costs, energy efficiency, and security [9]. It is especially in the latter five aspects that OCC plays a pivotal role. Unlike Radio Frequency (RF) waves, which can be detected from several meters away, OCC confines the possibility of detection to the area illuminated by the light source, providing greater confidentiality, particularly indoors. Similarly, OCC technology allows for cost reduction in terms of acquisition, mobility, and maintenance due to its straightforward operation and installation. Moreover, it avoids contributing to electromagnetic spectrum saturation and is more immune to transmitter interference issues. The use of cameras also significantly increases the number of transmitters in scenarios, effectively monitors their potential mobility, and achieves location precision of less than a centimeter [8]. The key requirements proposed for 6G, as well as the contribution of OCC technology to these objectives, are detailed in Table 1.

The primary applications of OCC in the context of 6G will be oriented toward intelligent transportation systems [17], indoor positioning systems [8], and sensor networks [4] due to the aforementioned aspects. They will also be instrumental in achieving ubiquitous connectivity thanks to their excellent performance in underwater environments [18]. In this realm, OCC will also enhance node mobility compared to widely used wired systems.

## 2. Discovery Algorithm

The developed algorithm carries out transmitter detection by considering the main spatial–temporal characteristics that allow distinguishing a pixel associated with a transmitter from any other pixel. The four aspects considered for detection are pixel brightness, the size of the illuminated region, luminosity variation, and the patterns that follow these luminosity changes. Due to the high number of tasks to be performed, the algorithm was implemented through four phases, which will be discussed later in this section.

Both the minimum luminosity presented by transmitter pixels in the ON intervals ($L_{min}$) and the maximum size of the luminous regions associated with transmitters (Maximum Region of Interest or $ROI_{max}$) follow a dependent relationship with the average luminosity of the image. To evaluate this dependency, Equations (Equation 3) and (Equation 4) have been followed. To obtain these equations, a large number of samples of the parameters to be related were taken in different scenarios, and a script was developed to find a polynomial expression that would relate them. Therefore, the algorithm takes into account that while the projection of the transmitter on the image sensor is less than one pixel according to the formulas seen in (Equation 1) and (2), the luminous energy can affect several pixels due to the PSF. On the other hand, luminosity variation is generally close to zero in the case of non-transmitter pixels, while it takes a high value in the case of transmitters, which constantly oscillate between the ON and OFF states.
(3)$$L_{\min}=\sum_{i=0}^{4}a_{i}\cdot x^{i}$$
where the coefficients are $a_{4}=7.331\cdot 10^{-7}$, $a_{3}=-3.561\cdot 10^{-4}$, $a_{2}=0.052$, $a_{1}=-2.297$, and $a_{0}=164.8$.
(4)$$ROI_{\max}=\sum_{i=0}^{3}b\cdot x^{i}$$
where the coefficients are $b_{3}=-9.148\cdot 10^{-6}$, $b_{2}=6.749\cdot 10^{-3}$, $b_{1}=-1.558$, and $b_{0}=142.1$.

In order to evaluate the patterns by which transmitter luminosity could vary, as well as the minimum number of images to study to detect at least one bit as 1, it was necessary to first define the data frame to work with. The chosen frame was designed and evaluated in [14], and it favors the punctual and brief transmission of data to enhance its use in sensor networks. As observed in Figure 2, it only works with one byte of payload, and the situation where there are more than four consecutive bits set to 1 in the header to facilitate their detection can occur.

To implement the algorithm, Python was chosen as the programming language due to its wide range of resources available. Libraries such as Open Source Computer Vision (OpenCV) and scikit-image were used for image processing, while NumPy stood out for data structure handling. Resources related to multiprocessing were especially crucial for executing multiple processes in parallel. These resources were used to work with the images as they were captured as well as to reduce the processing times for those more complex tasks. For the first objective, a Queue object was used, and for the second, a Pool object was employed, allowing the creation of a group of processes. Additionally, the map function was utilized to execute a function multiple times on a list of iterables. This latter method was only employed in Phase 2, which had the highest computational complexity.

A general outline of the four phases comprising the image can be observed in Figure 3. As depicted, the first phase involves image extraction. In this phase, the images associated with a 4-s video segment are extracted one by one, taking into account the frames per second (FPS) rate at which the video camera operates. As the images are extracted, they are converted to grayscale and added to the Queue object to be concurrently processed in the next phase. The algorithm does not work with the RGB components because it needs to operate independently of the transmitter’s working length.

In the second phase, the number of potential transmitters is streamlined based on parameters linked to luminosity and the size of illuminated regions. To accomplish this, the images associated with the first two seconds of video are analyzed, as they are considered sufficient to detect at least one bit set to 1 if a continuous frame transmission is performed, as shown in Figure 2. In this phase, the average luminosity of one of the images is obtained, and the luminosity and region size thresholds are determined based on that value. Subsequently, the luminosity threshold is applied to the image, resulting in a matrix of ones and zeros. Next, all pixels that have passed the threshold and are contiguous are grouped together, and the number of pixels in each group is counted. If the number of pixels is less than the calculated threshold, the position of the central pixel is stored in a list of potential candidates. The different processes within the Pool generate different lists of solutions based on the list of images they receive to which they apply the aforementioned tasks. Before these lists are passed to the next phase, duplicate elements are removed, and they are merged. This list contains the pixels that are potential transmitter candidates.

In the third phase, the determination is made regarding which pixels are definitively linked to optical signals from transmitters based on the resemblance of their luminosity variation with the defined frame structure. To accomplish this, the corresponding frame is acquired for every pixel added to the potential transmitter list. To achieve this, the luminosity value of each pixel in the grayscale-converted images associated with 4 s of video is considered. Three filters are applied to each frame of each pixel, and if they pass these filters, the pixels are considered transmitters. The first filter is the standard deviation of luminosity, the second is the number of rising edges, and the third is the duration of the second period of the frame. To pass the last two filters, the frame is converted to 1 s and 0 s by applying the luminosity threshold. Thus, to obtain the number of rising edges, the frame is subtracted from itself shifted one position, and the number of −1 s is counted. On the other hand, to determine the duration of the second period, the absolute difference between the position of the first rising edge and the first falling edge is calculated. The resulting list from this phase contains only the pixels associated with transmitters.

Finally, in the fourth phase, the list of pixels classified as transmitters in the previous phase is traversed to check if there are pixels that are very close to each other, because it is assumed that these pixels likely belong to the same transmitter. The limit used is half of the value calculated using Equation (Equation 4). In this assessment, the distance between pixels is considered as the sum of the distance along the x-axis and the y-axis.

## 3. Methodology

### 3.1. Experimental Setup

To evaluate the algorithm, an OCC system was implemented in different scenarios, and communications were recorded in these settings. Subsequently, the algorithm was adapted to extract images from the video sequences as if it were real time.

The OCC system was implemented using embedded systems. On one hand, four Arduino Nano 33 devices were used, to which various types of LEDs were connected as transmitters. These devices transmitted all possible frames consecutively and indefinitely. On the other hand, a Raspberry Pi v3 was used with a high-performance camera as the receiver. The recorded videos had a minimum duration that allowed the recording of all 256 possible transmitted frames (1 byte of payload). Due to the memory and processing capabilities of the Raspberry Pi, the algorithm was executed on the videos using a laptop and an external hard drive. The characteristics of the main elements used can be observed in Table 2.

The system was implemented in three scenarios: 1 indoor, which can be seen in Figure 4, and 2 outdoors, which can be seen in Figure 5 and Figure 6. Different wavelength LED devices (red, green, blue and white) were used in these scenarios to assess how the algorithm’s performance varied with respect to the emission spectrum. Regarding the distance, minimum distances were maintained to achieve sub-pixel projections as defined in Equations (Equation 1) and (2). Specifically, a link range of 32 m was used in the indoor scenario, whilst 8 m and around 70 m were used for the outdoor tests during the daytime and nighttime, respectively. To evaluate these scenarios under different lighting conditions, several videos were recorded in some of them by modifying the exposure time and sensitivity parameters. Specifically, 3 videos were recorded in the indoor scenario, 2 in the daytime outdoor scenario, and 1 in the nighttime outdoor scenario.

### 3.2. Procedures

To validate the algorithm, the following workflow was evaluated. The experiments started by recording a video lasting at least 4 s in MJPEG format with a frame rate of 30 frames per second, ensuring that at least two samples of each bit were captured. The acquired images were stored for offline processing, composing the videos in a designated directory. The transmitters appearing in the video were programmed following the packet structure depicted in Figure 2. Additionally, these transmitters were positioned at a distance ensuring sub-pixel projection, as determined by the formulas specified in Equations (Equation 1) and (2).

During the offline processing stage, the directory where the video were stored was configured in the main script along with the frames per second rate at which the videos were recorded and the index of the initial image from which detection is desired. This initial image index should be sufficiently small to guarantee the examination of at least 4 additional seconds of images in the video. After this initial setting, the discovery algorithm was executed with these configured parameters, proceeding to the detection and discovery stage and displaying the number of transmitters, their positions and information, among other metrics detailed in the following section. An illustrative diagram of the described process is presented in Figure 7.

### 3.3. Metrics

Two types of parameters were evaluated from the recorded data: those associated with the algorithm’s quality and those associated with the communication quality. To assess the former, the true positive rate, false positive rate, positive omission rate, and processing times were used. To obtain these parameters, various graphs were presented after each execution of the algorithm, such as their location in a video image, their optical signal, or the execution times of each phase. Additionally, scripts were also developed to obtain these data by clicking on the pixel of interest to determine the reasons for positive omissions.

In contrast, to assess the quality of the communications, the Signal-to-Noise Ratio (SNR) and Bit Error Rate (BER) were used. These parameters were obtained using the Gaussian Mixture Model, whose expressions can be seen in (Equation 5) and (Equation 6). In these equations, $\mu_{0}$ and $\sigma_{0}$ are the mean and standard deviation of the first Gaussian, $\mu_{1}$ and $\sigma_{1}$ are the mean and standard deviation of the second Gaussian, and α is the weighting or relative contribution of the first Gaussian to the SNR calculation. The first Gaussian represents the signal, and the second Gaussian represents the background noise.
(5)$$SNR=\frac{1}{2}\frac{|\mu_{1}-\mu_{0}{|}^{2}}{\alpha \sigma_{0}^{2}+\left(1-\alpha \right)\sigma_{1}^{2}}$$
(6)$$BER=\frac{1}{2}\mathrm{erfc}\left(\sqrt{\frac{SNR}{2}}\right)$$

## 4. Results

After analyzing multiple samples from each of the recorded videos, it was deduced that the best results were obtained in the nighttime scenario, followed by the indoor scenario, and finally by the daytime outdoor scenario. However, the detection rate does not follow a linear relationship with the ambient light level. This will be explained in the following paragraphs and can be observed in Table 3, where the detection percentages of each tested scenario are listed.

In most indoor cases, the undetected transmitter was the same and corresponded to the one with the highest luminous power: the green LED. This was due to its saturation of the image sensor, which prevented it from dropping low enough during the OFF intervals. Consequently, due to the threshold used, its frame was read as a consecutive sequence of bits set to 1. This occurred more frequently during the average exposure time because Equation (Equation 3) adjusted the threshold less effectively in this case. On the other hand, in the daytime indoor scenario, in a significant number of cases, half of the transmitters were not detected, which was also due to the differences in luminosity among them. Specifically, the blue LEDs had high luminous power, while the red ones had low power. As a result, the OFF intervals of the LEDs with higher power had the same power as the ON intervals of those with lower power. This caused only half of them to be detected with the same sensitivity threshold. In the remaining cases, a very small percentage of the total positive omissions occurred when one or two transmitters were not detected due to some moving obstructive element.

Regarding the false positive rate, the only scenario where they were detected was in the indoor scenario. In this scenario, there were almost as many false positives as true positives. However, upon examining the optical signals of these false positives, it was found that they were not false positives per se but rather reflections of the transmitters on reflective surfaces that were mistakenly identified as transmitters. For example, in Figure 8, the little differences between the data frame received directly from the transmitter (upper) and from a reflection point on the ground (lower) are shown with the most significant distinction being the level of received power. In many instances, information from the omitted transmitter was obtained through its reflection. Therefore, in practical applications, these reflections can be used to improve the result reliability and maintain link continuity.

With respect to processing times, the nighttime scenario again yielded the best results (average of 48 ms), which was followed by daytime outdoor (average of 600 ms) and indoor (average of 680 ms). The significant difference in processing times between the first scenario and the other two is due to the substantial difference in ambient light and reflective surfaces. A greater number of these parameters resulted in more candidates exiting the simplification phase of the algorithm, requiring a greater number of pattern search operations. The correlation phase was the heaviest part of the algorithm, accounting for 85% of the processing time in the worst scenarios.

As depicted in Figure 9, illustrating the boxplot of processing times across the various tested scenarios, there is greater variability in processing times during indoor scenarios in the simplification phase. This is attributed to the luminosity and reflections within the indoor setting. Similarly, the processing times for the correlation phase exhibit increased variability in daytime scenarios overall, attributed to the higher number of transmitter candidates, which is due to a greater presence of brightness.

Finally, concerning SNR and BER values, better results were obtained in lower ambient light conditions due to the reduced noise effect. This can be observed in Table 4, which displays a summary of the average SNR and BER for each scenario and wavelength used. These values were obtained while operating with a voltage of 5 volts and a current of 5 mA. Likewise, lower ambient light conditions resulted in less variation in these parameters over time for the same transmitter as well as a smaller difference in the parameter values between transmitters. The transmitter that achieved the best results was the green one, which was followed by the blue, white, and finally the red. The best results, associated with the green transmitter, reached SNR values around 32 dB. On the other hand, the worst case presented an SNR around 11 dB. Nonetheless, both boundaries correspond to BER values far below the Forward Error Correction limit (usually defined as 3.2 × 10^−3^) according to Equation (Equation 6).

## 5. Conclusions

In the present study, an algorithm for sub-pixel optical transmitter detection has been developed and experimentally validated within the framework of camera-based optical communications. The developed algorithm is divided into four main blocks: image acquisition, simplification of potential transmitters based on their luminosity and size of the luminous region, correlation of optical signal values with theoretically expected values according to the used frame, and elimination of neighboring pixels. To reduce algorithm execution times, multiprocessing functions were used in the first two mentioned blocks, and to facilitate code development, the programming was carried out leveraging the various libraries and modules of the Python language.

The results demonstrated improved algorithm performance and better communication quality in lower ambient light conditions and its stability. Additionally, this had a significant impact on processing times, which did not exceed an average of 1 s in any case. Furthermore, working with transmitters of the same type and wavelength also yielded better results, as most of the omitted positive rates were due to differences in luminous power between transmitters and the challenge of finding a value that met these requirements. Expressions (Equation 3) and (Equation 4) are considered to depend on the type of transmitter being used.

While the algorithm’s results are considered acceptable, the experimental part has limitations. There is a recognized necessity to carry out recordings in new and diverse scenarios with varied environmental conditions. These additional recordings would allow for a more precise adjustment of the polynomial functions defined in Section 2 of this paper. Therefore, it could yield more accurate results in different environments. In addition, a greater number of recordings would enable a more precise definition of favorable wavelengths based on the environment and the determination of maximum distances considering environmental conditions.

To the best of the authors’ knowledge, this is the only OCC discovery algorithm to encompass sub-pixel casuistry. It is true that there are other OCC discovery algorithms that rely on temporal components. For instance, ref. [19] utilizes M sequences, while ref. [20] develops an asynchronous scheme. However, these algorithms operate with transmitter arrays, resulting in much larger projections than sub-pixel. Moreover, they do not offer all the metrics used in this document for evaluating the algorithm, including execution times, percentage of false positives, SNR, etc. Therefore, no comparison has been made with other algorithms given the novelty of the case in this document and due to the scarcity of relevant data in the most related works.

To conclude, this algorithm holds great significance for the development and introduction of OCC technology, which may have a substantial impact on 6G technology as a supporting technology. It is well suited for export as a web service or as an intermediary component for converting data into other network technologies. The primary application of this technology will focus on sensor networks, where it can enhance aspects such as node density, costs, and energy efficiency. This will have far-reaching implications across sectors such as industry, healthcare, the IoT, and more.

## Figures and Tables

**Figure 1 sensors-24-03249-f001:**
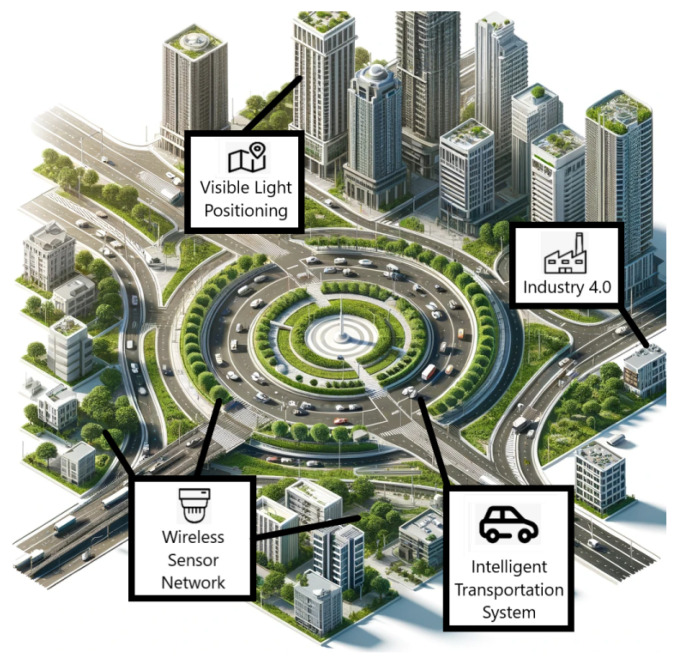
Concept on the use of OCC in Smart Cities.

**Figure 2 sensors-24-03249-f002:**
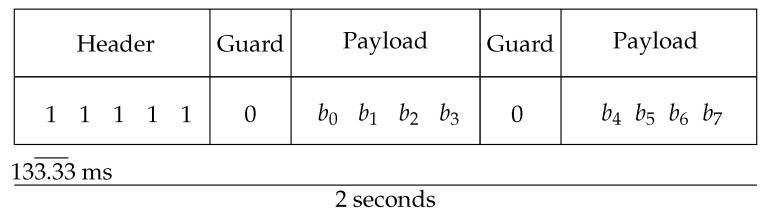
Communications data frame.

**Figure 3 sensors-24-03249-f003:**
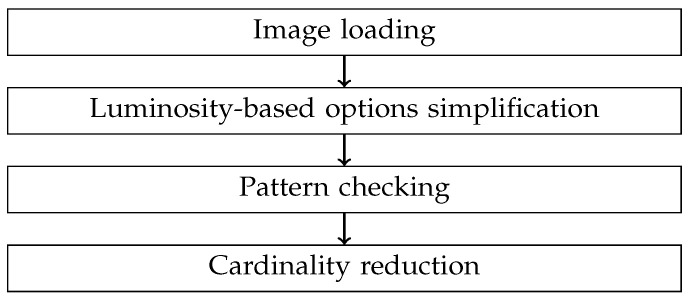
Algorithm phases.

**Figure 4 sensors-24-03249-f004:**
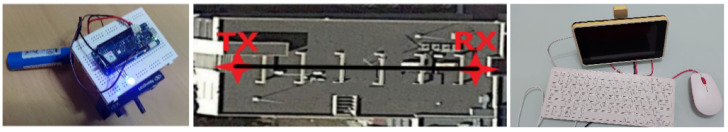
Indoor experimental setup. The resulting link range in the depicted corridor was 32 m.

**Figure 5 sensors-24-03249-f005:**
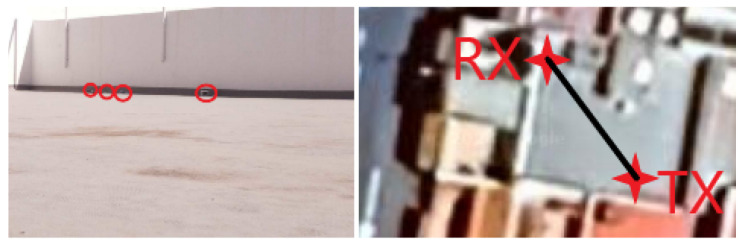
Outdoor daytime experimental setup. The resulting link range was 7 m.

**Figure 6 sensors-24-03249-f006:**
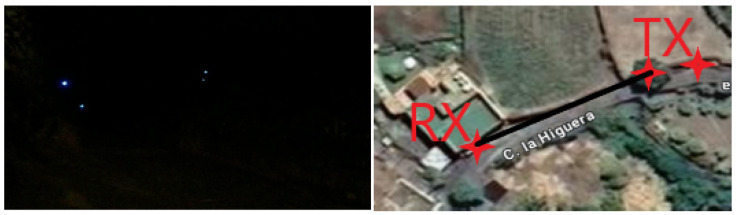
Nighttime experimental setup. The resulting link range was between 45 and 75 m.

**Figure 7 sensors-24-03249-f007:**
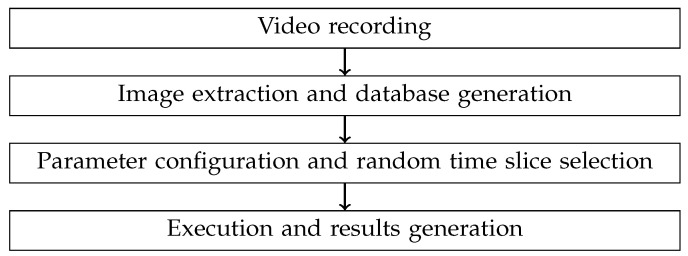
Procedure phases.

**Figure 8 sensors-24-03249-f008:**
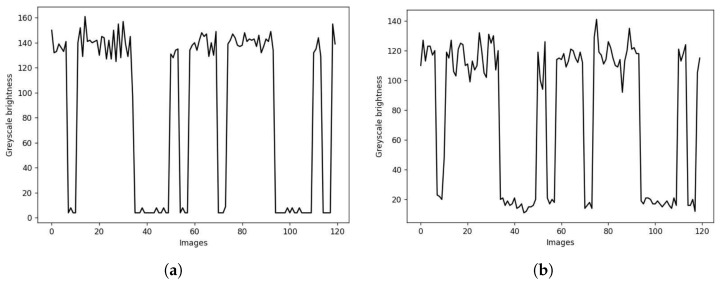
Data frame received directly from the transmitter (**a**) and from a reflection point on the ground (**b**).

**Figure 9 sensors-24-03249-f009:**
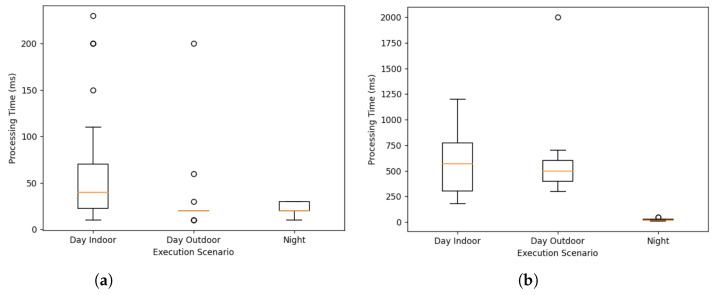
Boxplot of processing times by scenario for simplification (**a**) and correlation (**b**) phases.

**Table 1 sensors-24-03249-t001:** The 6G requirements and OCC support.

6G Requirement	Target Value	OCC Support
Peak Data Rate	Up to 1 Tbps	No
User-Experienced Data Rate	1 Gbps or higher	No
User Plane Latency	100 µs or lower	No
Mobility	Up to 1000 km/h	No
Connection Density	Up to $10^{7}$ devices/km^2^	Yes
Energy Efficiency	10–100 times better than 5G	Yes
Peak Spectral Efficiency	Three times higher than 5G	Yes
Area Traffic Capacity	Up to 1 Gbps/m^2^	Yes
Success probability (Reliability)	1–10^−7^	Yes
Signal Bandwidth	Up to 1 GHz or higher	No
Positioning Accuracy	Centimeter-level precision	Yes
Coverage	3D (terrestrial–satellite–aerial)	Yes
Timeliness	Real-time data emphasis	No
Security and Privacy	Ensured confidentiality, integrity, authentication	Yes
Capital and Operational Expenditure	Cost-effective networks	Yes

**Table 2 sensors-24-03249-t002:** Material characteristics.

Arduino Nano 33 IoT	
Microcontroller	Low-power ARM MCU SAMD21 Cortex^®^-M0+ 32-bit
Clock Speed	48 MHz
Flash Memory	256 Kb
Power Supply	3.3 V, 2.5 A
RGB LED	
Model	L-154A4SURKQBDZGW
Manufacturer	Kingbright
Size	5 × 8.6 mm
Wavelength	470 nm, 525 nm, 630 nm
Luminosity	150–300 mcd, 500–1000 mcd, 600–1300 mcd
White LED	
Model	SLD430WBD2PT3
Manufacturer	ROHM Semiconductor
Size	5 × 3 × 3 mm
Luminosity	1850 mcd
Raspberry Pi	
Model	Raspberry Pi 3 Model B
Manufacturer	Raspberry Pi Foundation
Operating System	Raspbian-32 bits
Processor	Broadcom BCM2837 SoC
Storage	1 GB of RAM
Power Supply	5 V, 2.5 A
Camera Module V2	
Model	IMX219—V2 module
Manufacturer	Raspberry Pi Foundation
Acquisition Mode	Rolling Shutter
Maximum Resolution	3280 × 2464
Used Resolution	1920 × 1080
Maximum Gain	16 B
Maximum Frame Rate	30 FPS
FOV	62.2 × 48.8

**Table 3 sensors-24-03249-t003:** Detection rates summary.

Scenario	Transmitters Detected		
	**2/4**	**3/4**	**4/4**
Indoor, low luminosity	0%	29%	71%
Indoor, medium luminosity	14%	57%	29%
Indoor, high luminosity	0%	21%	79%
Daytime outdoor, low luminosity	82%	18%	0%
Daytime outdoor, high luminosity	0%	45%	55%
Nighttime outdoor	0%	0%	100%

**Table 4 sensors-24-03249-t004:** Average SNR and BER in the different scenarios.

Scenario	LED Color	Average SNR (dB)	Average BER
Daytime Indoor	Red	20.6	2 × 10^−20^
	Blue	21	7.6 × 10^−22^
	Green	26.1	2.5 × 10^−90^
	White	20.8	6 × 10^−21^
Daytime Outdoor	Red	19	7 × 10^−16^
	Blue	24	2 × 10^−30^
	Green	-	-
	White	-	-
Nighttime Outdoor	Red	-	-
	Blue	27.2	3 × 10^−110^
	Green	-	-
	White	21.5	6.3 × 10^−33^

## Data Availability

Results can be provided upon request.

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
