# Peer review of "Cost-Effective Optical Wireless Sensor Networks: Enhancing Detection of Sub-Pixel Transmitters in Camera-Based Communications"

_sensors, 2024, doi:10.3390/s24103249_

Round 1
Reviewer 1 Report
Comments and Suggestions for Authors
Reviewer 2 Report
Comments and Suggestions for Authors
The paper presents a novel algorithmic approach to address the challenges associated with Optical Camera Communication (OCC) in Internet of Things (IoT) environments. The authors focus on enhancing the detection of sub-pixel transmitters without necessitating modifications to the camera or image sensor. The introduction of an efficient sub-pixel optical transmitter discovery algorithm demonstrates a methodical and technically sound approach. The reported detection rates of 100% in nighttime scenarios, 89% indoors, and 72% outdoors during daylight provide compelling evidence of the algorithm's efficacy across diverse environmental conditions.
My comments are listed as follows:
1. In the intro part, "However, none of them deal with sub-pixel scenarios and can be applied universally due to their focus on transmitters with defined size and shape or specific applications that are challenging to extrapolate." Is the proposed algrithm the only one to deal with sub-pixel scenarios? If not, compare with other sub-pixel optical transmitter discovery algorithm in the exsisting works.
2. In the result part, a table with detailed SNR and BER values should be supplemented that corresponds to the description in the last paragraph.
3. A comparison table should be supplemented to clarify the nolvety and advantages as compared with other algorithms in terms of scope of application range, cost and detection rates etc.
Round 2
Reviewer 1 Report
Comments and Suggestions for Authors
The author has addressed all the concerns I raised.